# Comparative analysis of the gut microbiota of sand fly vectors of zoonotic visceral leishmaniasis (ZVL) in Iran; host-environment interplay shapes diversity

**Fateh Karimian[1], Mona Koosha[2], Nayyereh Choubdar[2], Mohammad Ali Oshaghi[2]***

**1** Department of Parasitology, Pasteur Institute of Iran, Tehran, Iran, **2** Department of Medical Entomology and Vector Control, School of Public Health, Tehran University of Medical Sciences, Tehran, Iran

\* moshaghi@sina.tums.ac.ir

**Data Availability Statement:** All relevant data are within the manuscript and its Supporting Information files.

## Abstract

The development of *Leishmania* parasites within sand fly vectors occurs entirely in the insect gut lumen, in the presence of symbiotic and commensal bacteria. The impacts of host species and environment on the gut microbiome are currently poorly understood. We employed MiSeq sequencing of the V3-16S rRNA gene amplicons to characterize and compare the gut microbiota of field-collected populations of *Phlebotomus kandelakii*, *P. perfiliewi*, *P. alexandri*, and *P. major*, the primary or secondary vectors of zoonotic visceral leishmaniasis (ZVL) in three distinct regions of Iran where ZVL is endemic. In total, 160,550 quality-filtered reads of the V3 region yielded a total of 72 operational taxonomic units (OTUs), belonging to 23 phyla, 47 classes, 91 orders, 131 families, and 335 genera. More than 50% of the bacteria identified were Proteobacteria, followed by Firmicutes (22%), Deinococcus-Thermus (9%), Actinobacteria (6%), and Bacteroidetes (5%). The core microbiome was dominated by eight genera: *Acinetobacter*, *Streptococcus*, *Enterococcus*, *Staphylococcus*, *Bacillus*, *Propionibacterium*, *Kocuria*, and *Corynebacterium*. *Wolbachia* were found in *P. alexandri* and *P. perfiliewi*, while *Asaia sp.* was reported in *P. perfiliewi*. Substantial variations in the gut bacterial composition were found between geographically distinct populations of the same sand fly species, as well as between different species at the same location, suggesting that sand fly gut microbiota is shaped by both the host species and geographical location. *Phlebotomus kandelakii* and *P. perfiliewi* in the northwest, and *P. alexandri* in the south, the major ZVL vectors, harbor the highest bacterial diversity, suggesting a possible relationship between microbiome diversity and the capacity for parasite transmission. In addition, large numbers of gram-positive human or animal pathogens were found, suggesting that sand fly vectors of ZVL could pose a potential additional threat to livestock and humans in the region studied. The presence of *Bacillus subtilis*, *Enterobacter cloacae*, and *Asaia sp* suggests that these bacteria could be promising candidates for a paratransgenesis approach to the fight against *Leishmaniasis*.

**Funding:** Research was supported by Elite Researcher Grant Committee under award number 971055 from the National Institutes for Medical Research Development (NIMAD), Tehran, Iran to MAO. The funders had no role in study design, data collection and analysis, decision to publish, or preparation of the manuscript.

**Competing interests:** The authors have declared that no competing interests exist.

## Author summary

*Leishmania infantum*, a parasitic protozoan causing fatal visceral leishmaniasis, is transmitted to humans by several sand fly vectors. In this study, the microbiota within the midguts of *Phlebotomus kandelakii*, *P. perfiliewi*, *P. major* and *P. alexandri* was analyzed by 16S ribosomal DNA (rDNA) Miseq sequencing, revealing highly diverse community composition and abundance, from three diverse ecological and geographical regions of Iran. It appears that the gut microbiota is highly dynamic and controlled by multiple factors, including sand fly host and environment. Proteobacteria were the principal bacterial phylum isolated. High numbers of gram-positive human or animal pathogens were also found, suggesting that sand fly vectors of ZVL could pose a potential threat to livestock and human in the region. Furthermore, there was a positive correlation between vector capacity and bacterial diversities, where the weakest ZVL vector had the lowest diversity, whereas other, more efficient, vectors had higher diversity. This study showed that *Bacillus subtilis*, *Asaia sp.* and *Enterobacter cloacae* are possible candidates for a paratransgenic approach to reduce *Leishmania* transmission.

## Introduction

Sand flies accumulate *Leishmania* parasites by feeding on human and other animal reservoir hosts. The *Leishmania* parasite causes a spectrum of symptoms, including subclinical (inapparent), localized (skin lesion), and disseminated (cutaneous, mucocutaneous, and visceral) infections. Leishmaniasis is a parasitic disease that is reported in parts of southern Europe, the tropics, and subtropics, and is considered to be a neglected tropical disease (NTD). Out of the 20 NTDs ranked by the World Health Organization (WHO), the leishmaniases rank in the top three among those caused by protozoa [1].

Visceral Leishmaniasis (Kala-azar) (VL) is the deadly form of Leishmaniasis. In 2021 more than 90% of VL cases were reported from just 8 countries: Brazil, Eritrea, Ethiopia, India, Kenya, Somalia, South Sudan, and Sudan. An estimated annual incidence of VL in the world was over 30,000 new cases per year. In Iran, nearly 20 million people live in areas of endemic VL foci: with an estimated annual incidence of VL ranging from 100 to 300 cases. Visceral leishmaniasis is zoonotic in Iran and is caused by *Leishmania infantum*, and zoonotic visceral leishmaniasis (ZVL) is endemic in the northwestern, southern, and northeast regions of the country [2–5]. The causative agent of VL in different parts of Iran is transmitted by different species of sandflies: including *Phlebotomus kandelakii* (Shchurenkova, 1929), *P. tobbi* (Adler & Theodor, 1930), and *P. perfiliewi* (Parrot, 1930) in northwestern and northeastern Iran, and *P. major* (s.l.) (Annandale, 1910), *P. keshishiani* (Shchurenkova, 1936) and *P. alexandri* (Sinton, 1928) in southern parts [5–11].

Symbiotic and commensal microbes can confer numerous unfavorable, neutral, or beneficial effects on their arthropod hosts, and can play several roles in vector competence, nutritional adaptation, fitness, development, reproduction, defense against environmental stress, oviposition, egg hatching, larval survival, and immunity [12–20]. In sand flies, it has been demonstrated that these microbes play a critical role in *Leishmania* parasite growth, development, and vector competence [21–26]. The introduction of next-generation sequencing (NGS) technologies has permitted the rapid and more wide-ranging exploration of these microbial communities. NGS has provided a novel tool for the analysis of microbial communities infesting sand flies, including simultaneous and unbiased screening for various samples in a single sequencing run. Like many other arthropods, the advent of 16S rRNA profiling using NGS

sequencing methods has revealed complex microbiomes in sand flies. Recently the number of studies using NGS to investigate the microbial diversity and composition of sand flies has expanded [22, 27–30]. There are nine hyper-variable regions (V1-V9) of the bacterial 16S ribosomal RNA gene (16S) that can be targeted to identify bacterial taxa in 16S amplicon NGS studies, and regions V1-V3, V3-V5, V4-V5 have been the most targeted in microbiome studies.

We are developing a paratransgenic platform to control the transmission of *L. infantum* by the sand fly vectors [4]. Here, we assess the richness of gut bacterial species from four field-collected sand fly species. We investigated the effect of host and ecological variations on the bacterial diversity carried by sand flies in three endemic areas of ZVL in Iran, during the period of *L. infantum* transmission. Microbiome outlining of wild-caught sand flies will be of great help in exploring possible vector control candidates for a paratransgenic control approach.

## Methods

### Study areas

The present study was carried out in three endemic ZVL foci in northeastern (Bojnord in North Khorasan Province), northwestern (Meshkinshar in Ardabil Province), and southwest (Mamasani in Fras Province) regions of Iran (Fig 1).

North Khorasan Province (36˚37’–38˚17’N, 55˚53’–58˚20’E) is a mountainous region, 1070 meters above sea level and with an area of more than 28,400 km$^2$. The weather is hot (up to 32.4˚C) in summer and cold (minus 3.4˚C) in winter, with an average annual temperature of 13.2˚C. This region includes desert and mountainous areas and receives less than 250 mm rainfall annually. Ardabil Province (37˚04’–39˚65’N, 47˚40’– 48˚71’E) is a steppe region located 1490 meters above sea level with an area of more than 17,800 km$^2$. The weather is hot (up to 40˚C) in summer and cold (minus 20˚C) in winter, with an average annual temperature

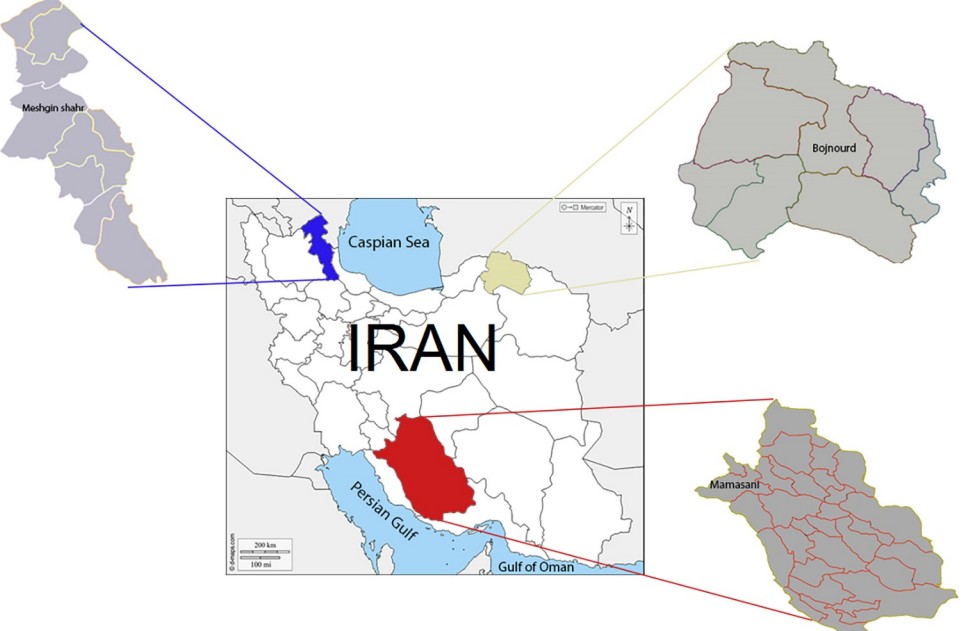

**Fig 1. Map of Iran and the sand fly collection sites.** Base layer is from (**https://commons.wikimedia.org/wiki/File: Map_of_Iran.png**).

of 9.5˚C. The warm season is short (mid-May to mid-September). The annual rainfall is approximately 325 mm, and the climate is warm and temperate, considered to be a local steppe climate. Fars Province (29˚37′–30″N, 52˚31′–54″E) is a steppe region located 1545 meters above sea level with an area of more than 122,608 km$^2$. The weather is hot (up to 29.2˚C) in summer and cold (minus 4.7˚C) in winter with an average annual temperature of 16.8˚C. The annual rainfall is approximately 100–800 mm.

## Sand fly collection and identification

Wild sand flies were collected from the study areas between 2015 and 2016, using various methods, including CDC light traps, and aspirators. Live sand flies were immediately transferred to the Insect Molecular Biology Laboratory, Department of Medical Entomology and Vector Control, School of Public Health, Tehran University of Medical Science, Tehran, Iran, under cold-chain conditions. Samples were washed first with sodium hypochlorite (bleach) 10%, followed by 70% ethanol for 3–5 min, then rinsed three times with sterile PBS and, finally, with double-distilled water. After the washing steps, sand fly guts were gently dissected under the stereomicroscope, using single use sterile insect needles. Before and between dissections, insect needles were sterilized by flaming. Dissections were done on sterilized single-use slide covers, and the heads and terminal abdominal segments of the collected sand flies were mounted with Pouri solution on glass slides for morphological identification to species level, using known morphological keys [31, 32]. A total of 48 female sand fly samples (6 pools, each one comprising 5–10 specimens), representing four species *P. kandelakii*, *P. perfiliewi*, *P. major*, and *P. alexandri* from the three locations (Meshkinshahr, Bojnord, Mamasani), were processed for microbiome identification (Table 1). To have similar samples, only female specimens with empty abdomens (either unfed or egg-laid blood-fed) were selected for microbiome processing.

## Identification of bacteria

Following sand fly species identification, microbiota definition was carried out only for gut specimens corresponding to known ZVL vectors. DNA was extracted from the homogenized gut pools using a DNA extraction kit (QiAamp DNA micro kit), following the manufacturer's recommended protocol. DNA was stored at -20˚C until used for sequencing.

The 16S rRNA gene hyper-variable V3 region was amplified by PCR using fusion degenerate primers 341F (5'-CCTACGGGAGGCAGCAG-3') and 518R (5'- ATTACCGCGGCTG CTGG -3'), and was sequenced on an Illumina Miseq platform. The amplified fragment was approximately 342 bp and raw data were screened and assembled by QIIME. The UCLUST method was used to cluster the sequences into Operational Taxonomic Units (OTUs) at an identity threshold of 97%. Each library pool was sequenced on a Junior+ System Genome Sequencer and then taxonomically assigned to bacterial genera by comparing and clustering

**Table 1. Details of the sand fly specimens processed for microbiome analysis.**

| District | Location | Coordinate | Species | Specimens tested |
|---|---|---|---|---|
| Meshkinshah | Northwestern | 37˚04'–39˚65'N, 47˚40'–48˚71'E | *P.kandelikii* | 8 |
| | | | *P. perfiliewi* | 9 |
| Bojnord | Northeastern | 36˚37'–38˚17'N, 55˚53'–58˚20'E | *P.kandelikii* | 8 |
| | | | *P.major* | 5 |
| | | | *P.alexandri* | 8 |
| Mamasani | Southwest | 30˚06′-30˚06′N, 51˚24′-51˚24′E | *P.alexandri* | 10 |

each sequence against the Greengenes database [33]. The sequence data obtained in this study have been submitted to the Genbank sequence read archive (SRA) under the following ID numbers: SRR19632069- SRR19632074.

As a negative control, we used the water from the final rinsing of the sand fly bodies and this was inoculated into Falcon tubes containing brain heart infusion (BHI) broth medium. To assess environmental contamination, the sand fly cuticles were used as an environmental control. These were removed from the sand fly carcass and subjected to DNA extraction by the phenol chloroform method; PCR amplification of 16s rRNA gene, as reported by Weisburg, (1991) produced a 1,500 bp fragment [34]. Thus, three no-template controls, PCR grade RNAse-free water, the final rinse water, and the sterilized cuticles were used to detect any bacterial and/or DNA contamination in the amplification reagents. Where the negative control was positive the specimen was eliminated from further analysis. Frequent changes of gloves were used to avoid RNAse-DNAse contamination. Surface sterilization of the workstation with bleach (10%) followed by alcohol (70%) was performed before and after each experiment. In addition, we used instruments that were autoclaved before and after handling each sample, and avoided talking, sneezing, and coughing, or touching the areas where DNA might be present.

## Data analysis

Cytoscape Software (http://www.cytoscape.org), a tool for visualizing complex networks between data, was used to visualize bacterial richness and shared bacteria in the three sand fly species through the network analysis [35]. Data, as CYS files containing vertices or nodes (representing symbiont bacteria and sand fly hosts) and edges (representing links), were submitted to Cytoscape software v.3.9.1. Bacterial and host nodes, as well as geographical region links, were colored to better demonstrate their interaction. Microsoft Excel, GraphPad Prism and Jvenn webtool software [36] were used for graphical representation.

## Result

The NGS method allowed the successful characterization of the microbiome of field-collected female sand fly guts. A 346 bp fragment of the hyper-variable V3 region of the 16S rRNA gene was PCR amplified from the genomic DNA pools (from sand fly gut), using specific universal primers, and was effectively sequenced using the Illumina-MiSeq platform. A total of 160,550 reads were generated after the removal of short reads, chimeras and the discard of spurious OTUs from all species analyzed. The average number of reads was 6,023 per female gut (Table 2 and Fig 1). The number of OTUs varied between species samples (minimum = 16,895, maximum = 43,948). *Phlebotomus alexandri* from the southwest of the country (Fars) was found to harbor the highest number of reads, followed, respectively, by *P. kandelakii* from

**Table 2. The number of bacterial reads of V3 region-16S rRNA gene of female sand fly ZVL vectors in Iran.** B: Bojnord in northeast, S: Mamasani in southwest, M: Meshkinshahr in northwest.

| Species-Sample | NO. of read | Reads per female sample | No. of Phylum | No. of Family | No. of Genus | No. of species |
|---|---|---|---|---|---|---|
| *P. alexandri*-B | 29242 | 3655 | 10 | 52 | 62 | 7 |
| *P. alexandri*-S | 43948 | 4394 | 17 | 82 | 174 | 26 |
| *P. kandelakii*-M | 30409 | 3801 | 14 | 79 | 145 | 25 |
| *P. kandelakii*-B | 16895 | 2112 | 14 | 69 | 120 | 15 |
| *P. major*-B | 21096 | 4219 | 9 | 55 | 82 | 9 |
| *P. perfiliewi*-M | 18960 | 2106 | 11 | 70 | 119 | 23 |

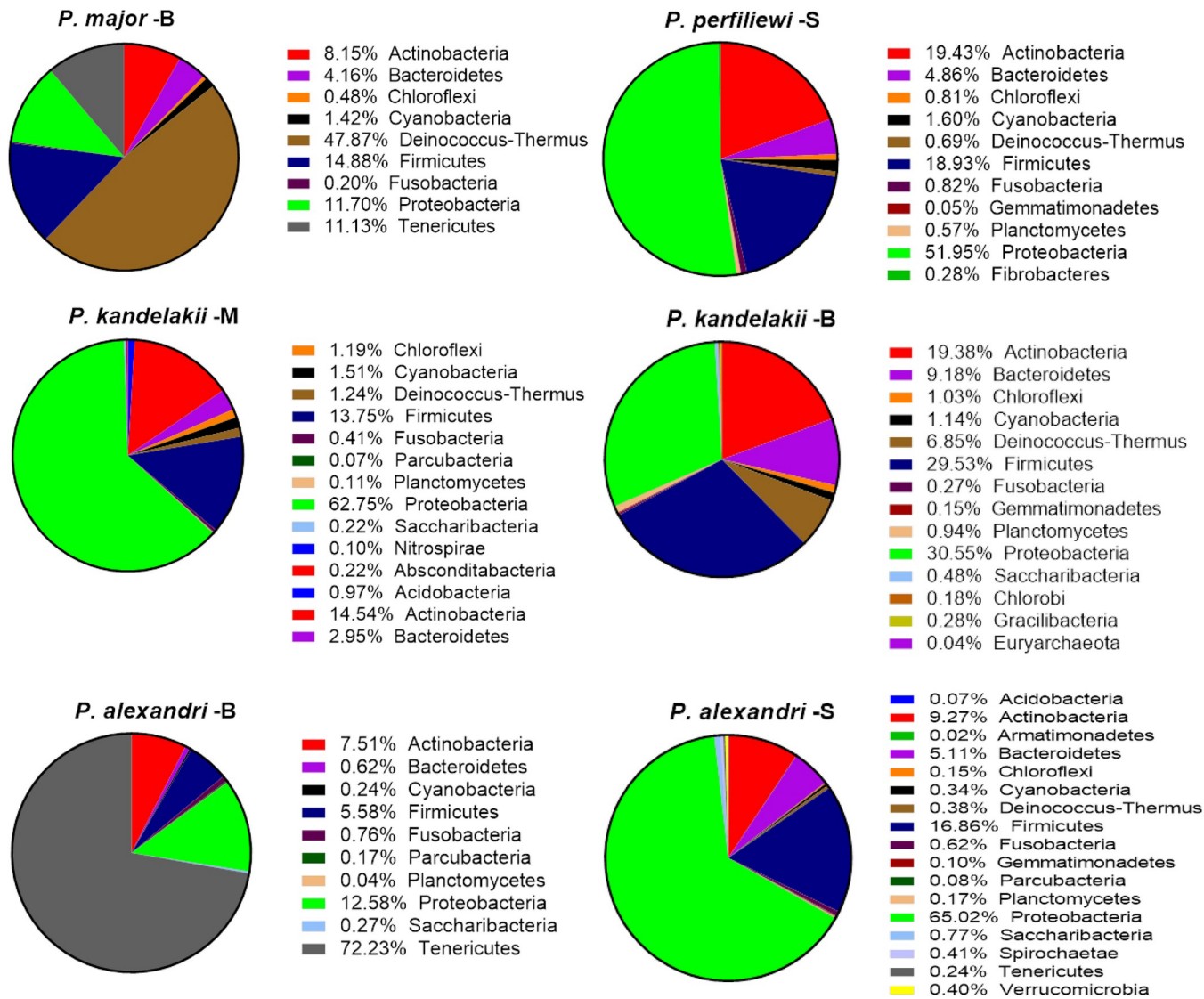

**Fig 2. Mean relative abundance of the bacteria at the Phylum level in sand fly species at different locations of Iran.** B: Bojnord in northeast, M: Meshkinshahr in northwest, S: Mamasani in southwest.

the northwest (Ardabil), *P. alexandri* and *P. major*, both from the northeast (Bojnord), *P. perfiliewi* from the northwest (Ardabil) and, finally, *P. kandelakii* from the northeast (Bojnord), which contains the lowest number of reads. These taxa spanned 23 phyla, 47 classes, 91 orders, 131 families, 335 genera, and 72 species.

In total, 160,550 quality-filtered reads of the V3 region of 16S rRNA gene were obtained and clustered into 335 operational taxonomic units (OTUs) at the genus level, with 97% similarity. In total, 23 bacterial phyla were associated with the gut microbiome. The phylum Proteobacteria makes up the highest number of reads in in the female guts. Proteobacteria, Firmicutes, Deinococcus-Thermus, Actinobacteria, and Bacteroidetes with, respectively, 51%, 22%, 9%, 6%, and 5%, were the most abundant phyla and were present in all sand fly species. The relative abundance of the bacterial phyla in different species and localities is summarized in Fig 2.

A total of 131 unique bacterial taxa families were detected in the species samples. We detected four families with high average relative abundances in sand fly samples: Anaplasmataceae (17%), Spiroplasmataceae (15%), Methylobacteriaceae (12%), and Thermaceae (0.8%). A Venn diagram analysis revealed that a subset of **19** bacterial taxa families were common across all different species and locations (Fig 3).

At the genus level, a total 335 genera were identified from the four sand fly species originating from three diverse regions of the country. There were 145 bacterial genera in *P. kandelakii* (Meshkinshahr, northwest), 62 in *P. alexandri* (Bojnord, northeast), 82 in *P. major* (Bojnord, northeast), 119 in *P. perfiliewi* (Meshkinshahr, northwest), 174 in *P. alexandri* (Mamasani, southwest) and 120 in *P. kandelakii* (Bojnord, northeast). Among these, 17 genera: *Spiroplasma*, *Pseudomonas*, *Acinetobacter*, *Tepidimonas*, *Sphingomonas*, *Wolbachia*, *Paracoccus*, *Methylobacterium*, *Streptococcus*, *Enterococcus*, *Staphylococcus*, *Pavimonas*, *Lactobacillus*, *Meiothermus*, *Propionibacterium*, *Kocuria*, and *Corynebacterium* were the most prevalent. The reproductive endosymbiont *Wolbachia* comprised 16% of the total reads and was recorded in *P. alexandri* from the southwest and *P. perfiliewi* from the northwest of the country. Also, we identified for the first time, *Asaia* sp. bacteria in the *P. perfiliewi* sandfly. The relative abundance of each genus is shown in Fig 4.

At the species level, a total of 72 species were identified from the four sand fly species originating from three diverse regions of the country. There were 26 bacterial species in *P. alexandri* (Mamasani, southwest), 25 in *P. kandelakii* (Meshkinshahr, northwest), 23 in *P. perfiliewi* (Meshkinshahr, northwest), 15 in *P. kandelakii* (Bojnord, northeast), 9 in *P. major* (Bojnord, northeast), and 7 in *P. alexandri* (Bojnord, northeast). *Bacillus subtilis* and *Pseudomonas aeruginosa* were found in all four sand fly species studied, while *Kocuria palustris*, *Aeromonas spp.*, and *Enterobacter cloacae* were found in three of the four sand fly species. The most frequently isolated bacteria in sand flies were *Wolbachia* spp., followed by *Pseudomonas aeruginosa*, *Lysinibacillus sphaericus*, *Kocuria palustris*, *Bacillus subtilis*, *Enterobacter cloacae*, *Streptococcus constellatus*, and *Bacillus licheniformis*. 53 out of the 72 bacterial species (73.6%) were found in either a single sand fly species or in one specific location. *Bacillus subtilis* was found in all the sand fly specimens, except for the northeast population of *P. kandelakii* (Fig 5).

Further analysis of sequence reads revealed that, on average, the microbiome of the four sand fly species is more associated with gram positive (57.7%) and pathogenic bacteria (69.15%) (*Wolbachia* was excluded from analysis): with 52 out of 72 species being known as human or animal pathogenic bacteria (S1 Table).

## Effect of ecological habitat on microbiome community

To study the effect of the sand fly's ecological habitat on the bacterial community of its gut, we compared Alpha-diversity indices (Shannon, Simpson-e) that describe the diversity of the microbial community of the same sand fly genus/species at different sampling locations. Here we compared the microbial communities of two populations of *P. kandelakii* (northwest versus northeast), and two populations of *P. alexandri* (northeast versus southwest). This analysis showed a considerable variation in the composition of the microbial community in sand flies collected from different locations. For *P. alexandri*, the diversity of the southwestern population was much richer than that of the northeastern one, and for *P. kandelakii*, the population diversity of the northwestern population was higher than that of the northeastern population. Nonetheless, the same sand fly species from different locations shared a few 'core' bacterial taxa; however, network analysis showed that the number of species-shared bacteria (n = 3–6, 10–15.8%) is much lower than the number of location-specific bacteria (Fig 6) in both the sand fly species analyzed.

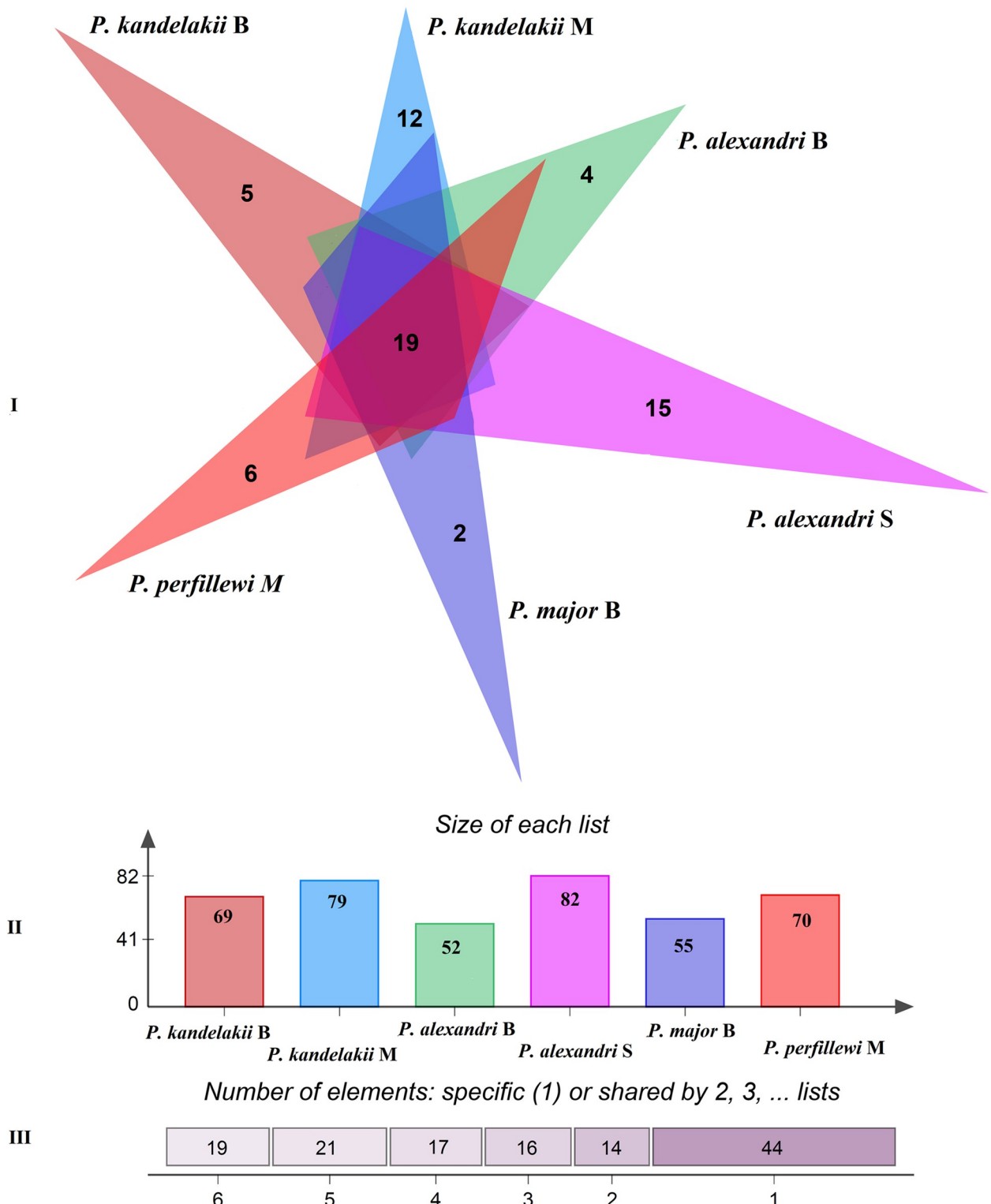

**Fig 3. Venn diagram of the bacterial families present in four sand fly species and their associated locations. I:** B: Bojnord in northeast, S: Mamasani in southwest, M: Meshkinshahr in northwest of Iran. The shared bacteria with less than 3 families are not shown. **II:** numbers of each family of bacteria in each sand fly population. **III:** the total number of bacterial families shared by the respective number of sandfly specimens. Venn diagram was constructed using the jvenn webtool [36].

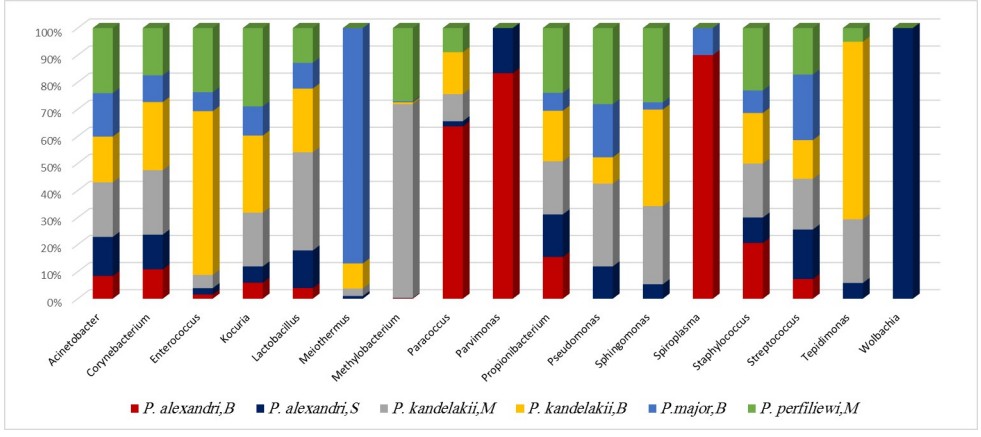

**Fig 4. Relative abundance of the common bacterial genera presents in four different sand fly ZVL vector species collected from three diverse ecological locations.** B: Bojnord in northeast, S: Mamasani in southwest, M: Meshkinshahr in northwest of Iran.

### Effect of host species on microbiome community

To study the effect of the host species on the bacterial community of the gut, we compared Alpha-diversity indices (Shannon, Simpson-e) that describe the diversity of the microbial community in different host species from the same sampling locations (sympatric species). Here we compared the gut microbial communities of *P. kandelakii* and *P. perfiliewi* from the northwest, and those of *P. alexandri*, *P. major*, and *P. kandelakii* species from the northeast of the country. This analysis showed that the diversity of the microbial communities differs between all four species; with the diversity being greatest in *P. kandelakii*, followed by those of *P. perfiliewi* and *P. major*, with the least diversity being observed in *P. alexandri*. Network analysis showed that the number of bacteria shared between sympatric species is much lower than species-specific bacteria (1 versus 5/7/13 and 10 versus 13/15) (Fig 7), indicating that host-specific factors influence the overall composition of the bacterial community. For example, there were no core bacteria shared between the three sympatric species (*P. alexandri*, *P. major*, and *P.kandelakii)* collected from the northeast, *B. subtilis*, *K. palustris* and *H. ganmani* shared only between *P. alexandri* and *P. major*, *P. major* and *P. kandelakii*, and *P. alexandri* and *P. kandelakii* respectively, and only 10 out of 38 (26.3%) bacterial species were shared between *P. kandelakii* and *P. perfiliewi* from the northwest.

### Discussion

This study provides evidence on the microbiome composition of the midgut of four Old World ZVL vectors, *P. kandelakii*, *P. perfiliewi*, *P. alexandri*, and *P. major*. Our results show that, in total, more than 51% of the bacteria identified belong to the phylum Proteobacteria, which is partly in accordance with the results (56.4%) from a previous culture-dependent study on three sand fly species, *P. major*, *P. kandelakii* and *P. halepensis* (Theodor, 1958) [4] from northern Iran, and several other Old World sand fly species (47%) [22, 28, 37–41]. Interestingly, the abundance of Proteobacteria phylum in New World (*Lutzomyia* sp.) sand fly species has been shown to be slightly higher (57–67.6%) than in Old World species [27, 30, 42–45]. Since members of the Proteobacteria phylum can fix atmospheric nitrogen and contribute to host sustenance [46], it is possible that this difference might reflect differing nutritional constraints between Old and New World sandfly species and /or environments.

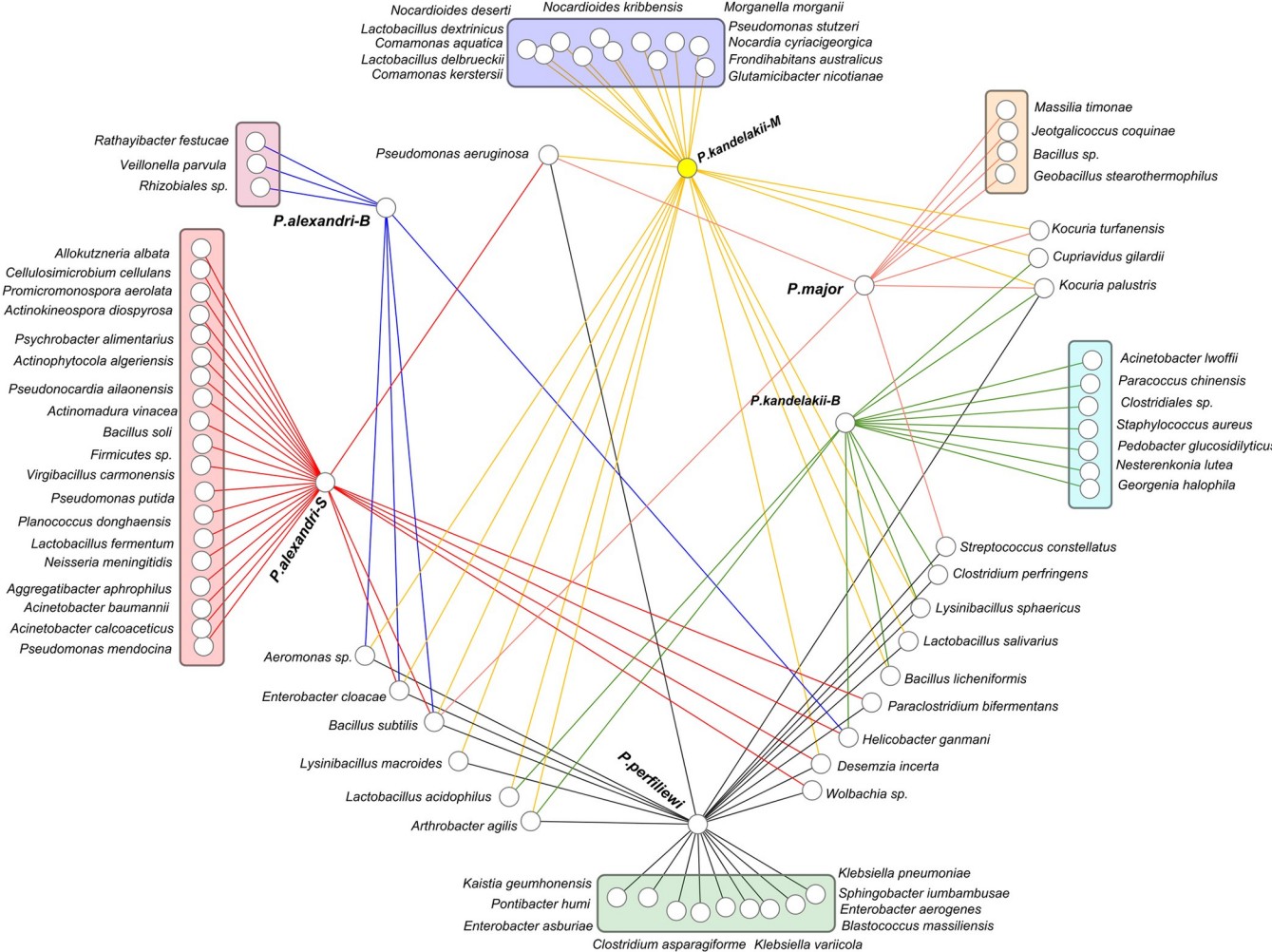

**Fig 5. Network analysis showing the shared and non-shared gut bacteria of four sand fly species collected from three diverse regions of Iran, revealed by NGS.** B: Bojnord in northeast, S: Mamasani in southwest, M: Meshkinshahr in northwest of Iran.

Our analysis also shows that the Firmicutes phylum is the second most abundant bacterial phylum in the four sand fly species in Iran, which agrees with previous studies [22]. Interestingly, however, the percentage abundance (22%) that we detect is lower than that previously reported for Old World sand fly species (39.8–41.42%), and more comparable with that previously found for New World species (23.9%) [22].

In the present study, four bacterial families were found with high average relative abundance in the four sand fly species: Anaplasmataceae (17%), Spiroplasmataceae (15%), Methylobacteriaceae (12%), and Thermaceae (0.8%). However, a meta-analysis study [22], previously showed that species of the Enterobacteriaceae family were the most prevalent (>60%) in both the New- and Old-World sand fly species, followed by those from the Moraxellaceae and Pseudomonadaceae families (<20%) and Xanthomonadaceae (<10%). Such differences in the gut microbiome composition might be the result of several factors, but in general, they could be explained by the phylogenetic relatedness of the sand flies and the diversity of their habitat.

Considerable numbers of pathogenic bacteria species were recorded in the four sand fly species, which might suggest that, as well as being vectors of ZVL, sand flies could also pose an additional threat to the health of livestock and humans. The risk of these pathogenic bacteria

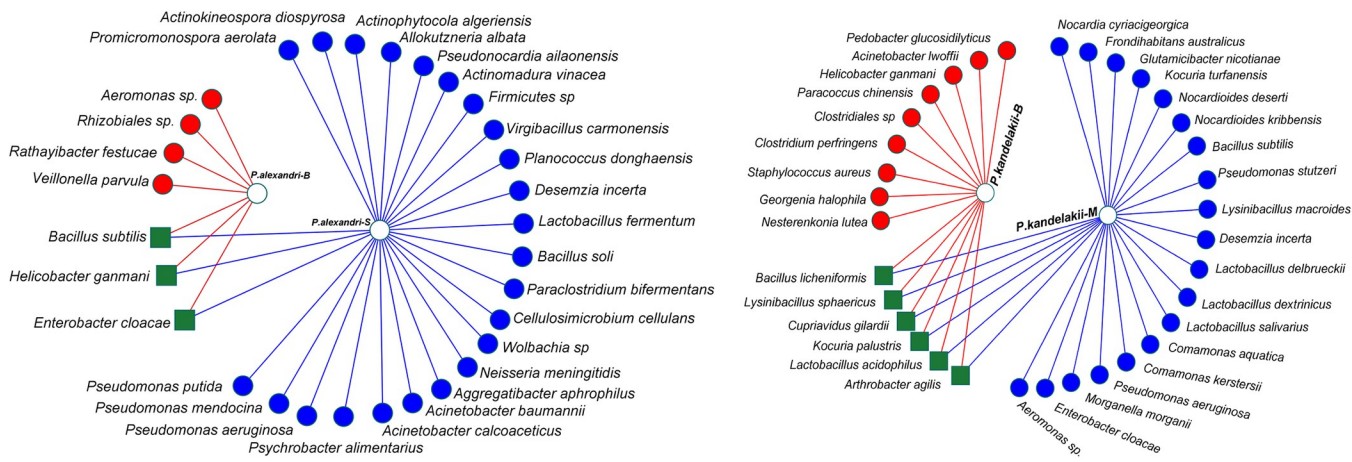

**Fig 6. Network analysis showing the shared (green squares) and non-shared (blue/red circles) gut bacteria from samples of the same sand fly species collected from two diverse regions of Iran.** Left panel: two populations of *P. alexandri* (B: northeast versus S: southwest), right panel: two populations of *P. kandelakii* (M: northwest versus B: northeast).

remains to be determined. Furthermore, the relative prevalence of putative pathogens was highly dependent on the sand fly species, and there was a positive correlation between vector capacity and prevalence of pathogenic bacteria. *Phlebotomus major*, which is known to be the weakest ZVL vector in Iran, carries the lowest rate of pathogenic bacteria (23.5%), whereas the other three species, which are more important ZVL vectors, carry much higher rates (66.3–88.5%) of pathogenic bacteria.

This study has shown that that the bacterial diversity in the gut microbiomes of *P. alexandri*-S (26), *P. kandelakii*-M (25), and *P. perfiliewi*-M (23) was significantly higher than in that of *P. major* (9). Accordingly, *P. alexandri*, *P. kandelakii* and *P. perfiliewi* are the main ZVL vectors in southern and northwestern parts of the country. Interestingly, in the northeastern area, where these sand fly species do not play a major role in ZVL transmission, their bacterial diversity decreased (to 15 for *P. kandelakii*-B and 7 for *P. alexandri*-B). The influence of the bacterial composition of the microbiome on the competence of insect vectors of parasitic diseases has already been confirmed in mosquitoes, sand flies, ticks, and tsetse flies [4, 23–24, 47–50]. For example, using antibiotics to disturb the gut microbiota of sand flies *P. duboscqi* (Neveu-Lemaire, 1906) and *Lutzomyia longipalpis* (Lutz & Neiva, 1912) correspondingly halted the

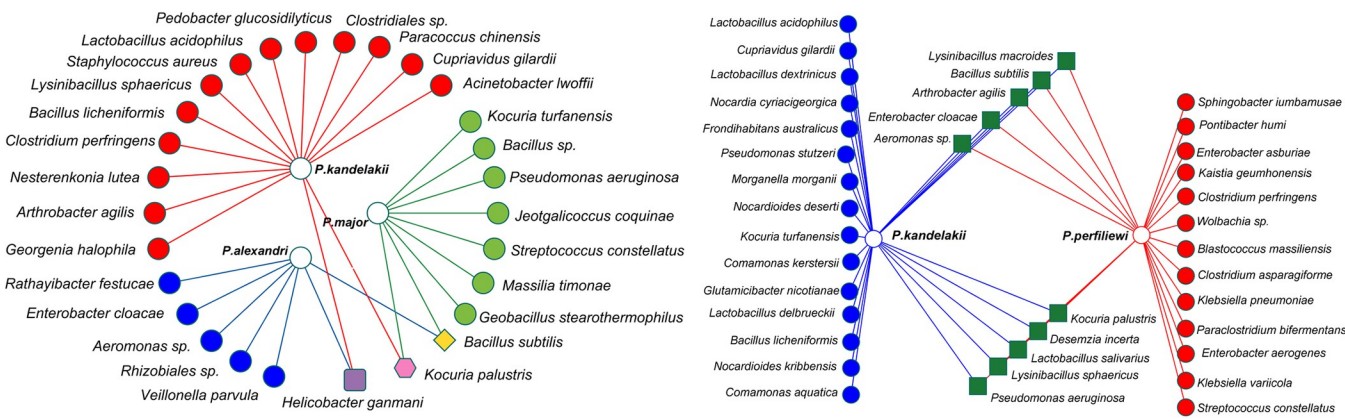

**Fig 7. Network analysis showing the shared (yellow square, purple square, pink hexagon, green squares) and non-shared (red, green, and blue circles) gut bacteria of sympatric sand fly species in northeast (left panel) and northwest (right panel) locations of Iran revealed by NGS.**

development and expansion of *Leishmania major* and *L. infantum* within the sand fly guts [23–24]. Thus, the interaction between the gut microbiome of the sand fly host and the *Leishmania* parasites appears to be beneficial for the parasites. In addition, studies in mosquitoes showed that some gut bacterial species can directly or indirectly reduce [51–57], or enhance [58–60], parasite transmission. Conversely, pathogens such as the malaria parasite and Zika, and Chikungunya viruses can shape the abundance and composition of the mosquito gut microbiome [12, 61–63]. However, less information is currently available on the influence of bacteria on the vectorial competence of sand flies, and it is essential to determine what bacterial species and by which mechanism(s) the bacterial microbiome may enhance or repress *Leishmania* development in the sand fly gut. It is worth mentioning that unfortunately in this work alterations of gut sand fly microbiota due to seasonal variations were not studied. Seasonal alterations could be related to an increase or decrease in pathogens transmission.

In this study we found some *Bacillus* species, including *B. soli*, *B. licheniformis*, and *B. subtilis*, in the four sand fly species, as expected, since the bacteria of this genus are found in almost all Old-World sand fly species [22]. *Bacillus subtilis* was found in all four sand fly species studied here (*P. kandelakii*, *P. alexandri*, *P. perfiliewi*, and *P. major)*, and it has previously been isolated from *P. major*, *P. halepensis*, *P. papatasi* (Scopoli, 1786) and *P. perniciosus* (Simic, 1932) [4, 22, 41]. Its frequent presence in several sand fly species and its being non-pathogenic, easily culturable and genetically malleable led to its consideration as a good bacterial candidate for paratransgenic approaches. Indeed, it has previously been used as a promising paratransgenic agent to impair parasite growth and reduce *Leishmania* transmission [22, 41, 64]. In addition to *B. subtilis*, we have isolated *Asaia sp.* and *Enterobacter cloacae* from the sand flies, both of which are known to be suitable paratransgenic agents, having previously been used to develop paratransgenic mosquitoes [57, 65–67].

Although it is well known that host species and ecological factors can have a strong impact on insect gut microbiota [68–70], the impact of these factors on sand fly microbiota remains poorly understood. The results of this study show that both environmental and host species identity can have a marked effect upon the microbial communities in the sand fly midgut, with distinct microbial communities being found in different populations of the same sand fly species. Also, we showed that the microbial communities of different sympatric species were distinct from each other. Thus, both host phylogeny and ecological factors can influence gut microbial composition and diversity, potentially impacting pathogen acquisition and transmission by the sand fly vectors. It is known that several factors can influence the composition of microbial communities, including host species, genetic background, blood-meal source, larval and adult environment, climate, temperature, humidity, site and season of collection, body size, sex, stage of development, infection with pathogens and other microbes, and previous exposure to insecticides [22, 48, 50, 63, 68–79]. These results highlight the need for further studies to decode the roles of ecological and host factors in determining the gut microbiome and, hence, the vector competence of different sand fly species.

## Conclusions

This is the first report of gut bacterial microbiome of wild-caught *P. kandelakii*, *P. perfiliewi*, *P. alexandri*, and *P. major* collected in three endemic areas for ZVL in Iran. Our results show the presence of several pathogenic bacterial species, suggesting that sand fly vectors of ZVL also could pose an additional potential threat to livestock and humans in the country. We also show that *Bacillus subtilis*, *Enterobacter cloacae*, and *Asaia sp.* are possible candidates for a paratransgenic approach to reduce *Leishmania* transmission. Further studies are needed to decode the role of the gut microbiome in the vector competence of different sand fly species.

## Supporting information

**S1 Table. Taxonomic, types of gram stain, characters, and the number of operational taxonomic units (OTUs) in sand fly female guts revealed by NGS.** P: pathogen, NP: non-pathogen, M: northwest, B: northeast, S: southwest.
(DOCX)

## Acknowledgments

Authors would like to express our deep and sincere gratitude to Dr Miranda Thomas from ICGEB for her insightful suggestions and careful editing of the manuscript.

## Author Contributions

**Conceptualization:** Fateh Karimian.

**Data curation:** Mohammad Ali Oshaghi.

**Formal analysis:** Fateh Karimian, Mona Koosha, Nayyereh Choubdar.

**Funding acquisition:** Mohammad Ali Oshaghi.

**Investigation:** Mohammad Ali Oshaghi.

**Methodology:** Fateh Karimian, Mona Koosha, Nayyereh Choubdar.

**Resources:** Mohammad Ali Oshaghi.

**Software:** Fateh Karimian.

**Supervision:** Mohammad Ali Oshaghi.

**Validation:** Mohammad Ali Oshaghi.

**Visualization:** Fateh Karimian.

**Writing – original draft:** Fateh Karimian.

**Writing – review & editing:** Mohammad Ali Oshaghi.

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
