## [Decision Letter · Decision Letter 0]

27 May 2022

Dear Prof Oshaghi,

Thank you very much for submitting your manuscript "Comparative analysis of the gut microbiota of sand fly vectors of zoonotic visceral leishmaniasis (ZVL) in Iran; host-environment interplay shapes diversity" for consideration at PLOS Neglected Tropical Diseases. As with all papers reviewed by the journal, your manuscript was reviewed by members of the editorial board and by several independent reviewers. The reviewers appreciated the attention to an important topic. Based on the reviews, we are likely to accept this manuscript for publication, providing that you modify the manuscript according to the review recommendations. 

Your manuscript has undergone careful revision by two reviews, and we are pleased to inform you that both have suggested that your manuscript be accepted with minor revisions and modification to one figure to make it more comprehensible.

Additionally please note that “All data is presented in the manuscript” is not an appropriate data availability statement for a microbiome profiling experiment. 16S sequences should be submitted to an appropriate repository (Genbank SRA) and accessions listed in the data availability section.

Sincerely,

Matthew Brian Rogers, Ph.D.

Associate Editor

Hans-Peter Fuehrer

Deputy Editor

Your manuscript has undergone careful revision by two reviews, and we are pleased to inform you that both have suggested that your manuscript be accepted with minor revisions and modification to one figure to make it more comprehensible.

Additionally please note that “All data is presented in the manuscript” is not an appropriate data availability statement for a microbiome profiling experiment. 16S sequences should be submitted to an appropriate repository (Genbank SRA) and accessions listed in the data availability section.

Reviewer's Responses to Questions

**Key Review Criteria Required for Acceptance?**

**Methods**

-Are the objectives of the study clearly articulated with a clear testable hypothesis stated?

-Is the study design appropriate to address the stated objectives?

-Is the population clearly described and appropriate for the hypothesis being tested?

-Is the sample size sufficient to ensure adequate power to address the hypothesis being tested?

-Were correct statistical analysis used to support conclusions?

-Are there concerns about ethical or regulatory requirements being met?

Reviewer #1: yes

Reviewer #2: The experimental design employed in this study was adequate to study objectives, the comparison of the microbiota composition of four sand flies species from three different geographic locations in Iran, using 16S RNA gene sequencing technique. My critical observation is about the number of specimens analyzed, been just 48 sand flies female samples. A higher number of insects could produce more robust results.

**Results**

-Does the analysis presented match the analysis plan?

-Are the results clearly and completely presented?

-Are the figures (Tables, Images) of sufficient quality for clarity?

Reviewer #1: figures need some updates

Reviewer #2: The results are clearly described, and the figures are clear and easy to understand, except for the figure 3- I, the Venn diagram. I consider this figure confused with many numbers inside and outside the diagram. This visual pollution makes it difficult the result understanding showed in this figure. I suggest redoing this figure.

**Conclusions**

-Are the conclusions supported by the data presented?

-Are the limitations of analysis clearly described?

-Do the authors discuss how these data can be helpful to advance our understanding of the topic under study?

-Is public health relevance addressed?

Reviewer #1: yes

Reviewer #2: The authors conclusions are adequate to results obtained, except to suggestion to use the Asaia bacteria in paratransgeneis approaches. Since it was only found in one species. Bacteria with greater distribution, present in several species, as seen with Bacillus subtilis, present in all species studied, present a greater potential for use in paratransgenic approaches.

**Editorial and Data Presentation Modifications?**

Reviewer #1: (No Response)

Reviewer #2: I suggest minor changes to the article. First change the Venn diagram figure, so that it is more understandable. Second, to improve the argument to suggest that despite having only been identified in a single species, the authors suggest that Asaia would be a good candidate for paratransgenic modifications.

**Summary and General Comments**

Reviewer #1: The paper “Comparative analysis of the gut microbiota of sand fly vectors of zoonotic visceral leishmaniasis (ZVL) in Iran; host-environment interplay shapes diversity” by Prof Mohammad Ali Oshaghi and colleagues is a well-written and welcomed contribution to the field. I only have a few comments and those mainly revolve around the figures and tables that should be made clearer.

Lines 89-91: what does “economical” mean in this context?

Line 133: not clear what you are dissecting.

Line 183: 160550 -> 160,500

Line 259: be more precise in what is “much much more”, a number or so

Fig. 2.: change so that the same phylum has the same color in all diagrams, now e.g. Cyanobacteria has six different colors, which makes it difficult to compare

Fig. 4.: put genera in alphabetical order

Fig. 6.: not clear what red and blue means, what is yellow?

Fig. 7.: explain purple, pink, yellow, and other colors and also the different geometrical forms

Table 1.: check space between genus and species, here and in the rest of the manuscript 

Table 2.: singular and plural for taxonomic levels should be adjusted

Table 3.: put as supplementary data and put species in alphabetical order

Reviewer #2: In this descriptive article, the authors compare the gut bacterial microbiota composition from sand flies of four different species and three geographic locations in Iran. Using 16 S high throughput DNA sequencing technique the authors observed that the bacterial microbiota composition was related to both, sand fly specie and the environment. Sympatric species present different bacterial species in their microbiota. An interesting observation was the relationship between the gut microbiota diversity and the sand fly vectorial capacities, where species presenting more diverse bacterial microbiota composition were the species with higher Leishmania transmission abilities. The identification of several pathogenic bacteria for human and animals in the sand flies guts also suggested a potential role sand flies in the transmission of the bacteria pathogens. Unfortunately in this work alterations of gut sand flies microbiota due to seasonal variations were not studied. Seasonal alterations could be related to a increase or decrease in pathogens transmission. The acknowledgment about the relationship among the vector, microbiota and pathogens can be used to develop mechanisms to control or eradicate the dissemination of several vector borne diseases. The authors suggest the use of Bacillus subtilis, Enterobacter cloacae and Asaia, as candidates for a paratransgenesis approach to the fight against Leishmaniasis. But from these three putative candidates, the only one found in all four sand flies especies studyed was B. subtilis. Commensals present on vectors species are better candidates for paratransgenesis approaches, than others identified in only one specie as was Asaia in this work.

PLOS authors have the option to publish the peer review history of their article (what does this mean?). If published, this will include your full peer review and any attached files.

Reviewer #1: No

Reviewer #2: No

Figure Files:

Data Requirements:

Reproducibility:

References

---

## [Editor Report · Decision Letter 1]

26 Jun 2022

Dear Prof Oshaghi,

We are pleased to inform you that your manuscript 'Comparative analysis of the gut microbiota of sand fly vectors of zoonotic visceral leishmaniasis (ZVL) in Iran; host-environment interplay shapes diversity' has been provisionally accepted for publication in PLOS Neglected Tropical Diseases.

Best regards,

Matthew Brian Rogers, Ph.D.

Associate Editor

Hans-Peter Fuehrer

Deputy Editor

Thank-you for your re-submission of this manuscript to PLOS Neglected Tropical Diseases. Based on your responses to the reviewers, and associated modifications to the text and figures your paper can now be accepted without further revisions.

---

## [Editor Report · Acceptance letter]

15 Jul 2022

Dear Prof Oshaghi,

We are delighted to inform you that your manuscript, "Comparative analysis of the gut microbiota of sand fly vectors of zoonotic visceral leishmaniasis (ZVL) in Iran; host-environment interplay shapes diversity," has been formally accepted for publication in PLOS Neglected Tropical Diseases.

Best regards,

Shaden Kamhawi

co-Editor-in-Chief

Paul Brindley

co-Editor-in-Chief
